# Research on a Commercial Building Space Traffic Flow Design Based on Post-Occupancy Evaluation

Yitong Zhu [1], Wenzhen Huang [1] and Linhui Hu [1,2,*]

1   School of Art and Design, Guangdong University of Technology, Guangzhou 510000, China; 2112017048@mail2.gdut.edu.cn (Y.Z.); 2112017009@mail2.gdut.edu.cn (W.H.)
2   Faculty of Innovation and Design, City University of Macau, Macau 999078, China
*   Correspondence: hlh@gdut.edu.cn

**Abstract:** The aim of traffic flow design in commercial building space planning is to create a comfortable, easily accessible, and identifiable spatial structure in a complex and changeable business environment. However, modern commercial building spaces often appear to be inconsistent with the usage requirements after they are put into use. To understand the real experience evaluation of users, this study selects the commercial project "Guangzhou Nanpu Times Commercial Building Space" as its subject. Based on post-occupancy evaluation (POE), this study uses the analytic hierarchy process (AHP) to construct an evaluation index system of commercial building space traffic flow. The weights and satisfaction scores of the indexes are calculated through expert interviews and questionnaires to analyze the evaluation and reasons for the commercial building space traffic flow design after using the facilities. We thereby obtain the design points of commercial building space traffic flow to provide a scientific, reasonable, and practical basis for the subsequent design of the same type of traffic flow to drive the prosperity and development of commerce.

**Keywords:** commercial building space; traffic flow design; post-occupancy evaluation (POE); analytic hierarchy process (AHP); satisfaction evaluation



## 1. Introduction

With the continuous improvement of people's living standards, urban commercial trade activities are increasingly diversified and complicated. As a component of trade activities, commercial building space design is developing towards the characteristics of service diversity, functional ambiguity, thematization, and large volume [1]. These characteristics also bring challenges to commercial building space traffic flow design. Commercial building space refers to a commercial aggregate in which a variety of retail stores and service facilities are concentrated in a building or an area to provide comprehensive services to consumers [2]. However, the traffic flow design is a critical component in commercial space design. User behavior does not simply occur in the space but is induced by the space type [3]. When consumers have a weak sense of position in commercial building space, it will significantly affect purchasing behavior, thus directly affecting the commercial space's use efficiency and economic benefits [4]. Therefore, in terms of design, traffic flow reflects the interactive relationship between commercial building space format and users, whose organization is based on creating a comfortable, accessible, and identifiable spatial structure in the complex and changeable commercial building space. Some scholars have identified existing problems with traffic flow. For example, Zhenmei Cui [5] summarized the main problems of traffic flow, including confusing layout, poor identifiability, and lack of features and themes; Qing Chen [6] stated that the design of traffic flow is short on variation, dimensionless in design, inflexible, etc. We consulted seven leading research scholars and planners of commercial spaces and found that most commercial building space traffic flow design was not well evaluated after use. This state of affairs may be due to various

reasons, e.g., the lack of organic connections within the overall structure and fragmentation of commercial building space [7], missing guidance identification information, etc. As the traffic flow reflects, the fit between the spatial pattern and the users' needs should be designed based on people-oriented principles [8]. Therefore, an evaluation system and information feedback method centered on users' needs may represent an essential step in traffic flow design.

By investigating the current situation of the design and use of commercial building space traffic flow in Guangzhou Nanpu Times, we analyze users' psychological and behavioral needs in commercial buildings and then grasp the characteristics of users' psychological and behavioral activities. Based on post-occupancy evaluation (POE), a commercial space traffic flow evaluation index system is constructed through the analytic hierarchy process (AHP), and a systematic and scientific evaluation of the commercial building space traffic flow design of Guangzhou Nanpu Times is carried out based on the feedback of the users. It aims to discover successes and shortcomings, provide valuable information for designers, analyze the problems in the design and use, and put forward reasonable and implementable suggestions. It is believed that our results can help us to better meet the requirements of consumers and provide a scientific, rational, and practical basis for commercial building space traffic flow design.

## 2. Literature Review

### 2.1. Commercial Building Space Traffic Flow

Traffic flow is the behavioral trajectory of people and goods flowing inside a building. It is also a reflection of the functional relationship of the building [9]. The traffic flow design is the link between the available units so that the commercial building space constitutes a complete and orderly whole [10]. The purpose of commercial building space traffic flow design is to ensure coordination between space organization, people, and things; to ensure the smooth flow of people and logistics; to avoid unnecessary crosses and interferences. In architectural design, the convenience and smoothness of the traffic flow indicate the rationality of functional partitioning, and the overall traffic flow should be balanced and predictable [11]. In addition, the designer should plan and operate a dynamic space that makes the flows of people, logistics, and vehicles meet the functional requirements while handling the basic functions of the building in a logical way [12] and maintaining a certain aesthetic effect.

Traffic flow design is an important topic that exists in commercial spaces. Scholars have conducted research on this topic from different perspectives, such as summarizing spatial organization patterns and emphasizing the importance of traffic flow to a spatial organization [13], improving spatial traffic flow planning through spatial syntax [14,15], and solving the problem of human circulation through spatial planning [16,17]. However, it neglects to understand the user's needs and psychological feelings and obtain measures for improvement from post-use evaluation. This study starts from the factors influencing the design of the traffic flow, scientifically and rigorously analyses the current problems of commercial building traffic flow design, solves the problems from the perspective of user psychology and behavior, and proposes design strategies.

### 2.2. Theoretical Framework

Post-occupancy evaluation (POE) is a study of the built environment developed in the 1960s from the field of environmental psychology, which refers to the systematic evaluation of facilities designed and used from the users' perspective [18]. By the 1980s, POE theory and methods had matured and were being applied in a variety of fields [19], with a focus on users and their needs, providing a solid basis for future design through in-depth analysis of the impact and operation of facilities once they are in use [20]. Specifically, POE refers to the use of questionnaires and expert interviews to verify the behavior and psychology of the users of a facility after it has been put into use. After scientific data analysis and aggregation, the users' evaluation of the facility is understood [21].

The American scholar W.F.E Preiser wrote the book Post Occupancy Evaluation in 1988, arguing that the beneficial factors of POE for the construction industry can be divided into short, medium, and long term, which shows its far-reaching influence on the construction industry [20]. In 2007, A. Leaman and B. Bordass combined the POE method with the building use research method, and as a result, the research became more diverse [22]. In 2010, these two scholars refined the principles and practices of POE methods for evaluating buildings, exploring the possibility of applying POE methods for buildings to other fields and laying the foundation for research in other fields [23].

When searching on Web of Science, Science Direct, and CNKI, it was found that the research results of POE evaluation methods in the fields of residential communities [24–26], schools [27–29], office buildings [30–32], urban parks [33,34], and urban leisure squares [35,36] have been significant. This theory can help us to understand the needs of users and improve the quality of project design and planning problems in planning and construction. Only a few scholars have focused on commercial building space, such as Meiqi Dong [37], who conducted a comparative case study of the use of four large commercial buildings in Shanghai's built environment. The scholar Linlin Qian [38] applied POE to the commercial space of the Beijing underground, analyzed the outstanding problems in the commercial development of the underground, and put forward corresponding suggestions for optimizing its design. It can be found that most of these scholars explore the study from the commercial building space as a whole, ignoring the issue of commercial building space subdivision. Few scholars have focused on studying commercial building space traffic flow design. By introducing POE into the design of commercial space, it is possible to understand the extent to which the commercial space traffic flow design meets the needs of the users from the perspective of the users.

To this end, this study takes the commercial project "Guangzhou Nanpu Times Commercial Building Space" as the subject. The project was put into use after construction was completed in 2019. After more than two years of operation, some shop tenants and customers have often reflected that there are limitations to the internal traffic flow. To improve the rationality of the commercial building space traffic flow and improve the experience evaluation of stakeholders, firstly, based on the POE, AHP is used to construct a commercial space traffic flow evaluation index system. Secondly, through expert interviews and questionnaires, users' satisfaction evaluation with the indexes after using the facilities is evaluated. Furthermore, we calculate the weight of each index and the satisfaction scores. Finally, the post-use evaluation and causes of the commercial building space traffic flow design are analyzed to derive the main points of the commercial building space traffic flow design. The analytic hierarchy process (AHP), first proposed by Saaty in 1977, is a multi-criteria decision-making method that obtains the weights of evaluation indexes relative to target values by comparing the importance of influencing factors in a hierarchical model with each other and ranking them hierarchically [39]. The evaluation indexes are the expression of the characteristics of a certain aspect of the evaluation object and its quantity, reflecting both the concept and nature of the characteristics of a certain aspect of the evaluation object and the number of the evaluation object's properties, with the dual role of qualitative and quantitative understanding of the evaluation object [40]. When it comes to commercial building traffic flow design evaluation, the evaluation indexes are the factor decomposition of relevant content for commercial building traffic flow design. At the same time, the index system is the collection of evaluation indexes and reflects the integration of information systems in all aspects of commercial building traffic flow design. A clear and scientific set of evaluation indexes is necessary to establish a commercial building traffic flow design evaluation index system. The evaluation indexes and their weights constitute the core part of the evaluation index system, which is the decisive factor in verifying evaluation results' accuracy and scientificity. Therefore, establishing a rigorous and scientific evaluation index system should follow the basic principles of wholeness, hierarchy, and scientificity [41,42].

### 3. Methodology

*3.1. Project Overview*

Nanpu Times is located on Nanpu Island in Panyu District, Guangzhou (Figure 1) and is funded by Guangzhou Ronghai Property Leasing Company Limited with a total construction area of 82,113 square meters and a total area of 16,025 square meters (Figures 2 and 3).

*3.2. Traffic Flow Design and Evaluation Target Setting*

Based on interviews and field research, the evaluation objectives are as follows:

1.  The current use of traffic flow design in the Nanpu Times is summarized through field research.
2.  Through expert interviews combined with the AHP method, the individual evaluation indexes of the traffic flow design are determined. Additionally, through matrix scoring and quantitative analysis, the weight values of each index are determined.
3.  The questionnaire survey of POE evaluation collects users' satisfaction with each index in the Nanpu Times traffic flow design. We calculate the relative importance by combining the index weight values to evaluate the overall traffic flow design, which will provide a reference basis for the future design and renovation of such space traffic flow.
4.  From the evaluation results, we not only analyze the problems and shortcomings of the Nanpu Times traffic flow but also suggest improvements.

**Figure 1.** Nanpu Times project location.

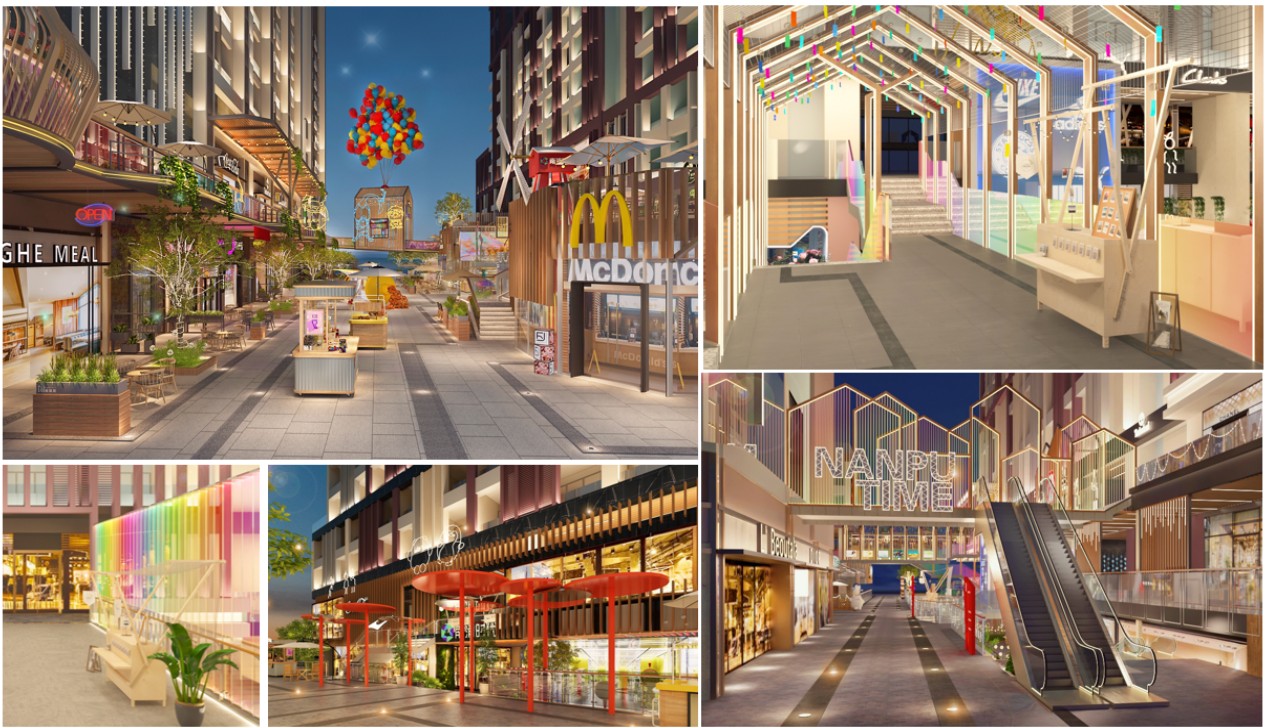

**Figure 2.** Overall plan of Nanpu Times.

**Figure 3.** Partial design effect of Nanpu Times.

### 3.3. Traffic Flow Design and Comprehensive Evaluation Method Design of Satisfaction

Based on POE, this study mainly adopts AHP, combined with expert consultation and scoring, to screen the evaluation indexes. Through pairwise comparison, we score the elements of each level, judge the relative importance, form the judgment matrix, and finally determine the significance of each factor through calculations and assign the corresponding proportion [39]. According to the user groups, including residents in the neighborhood of the commercial space, mobile people in the Guangzhou municipal district, staff providing services, consumers, etc., the Likert scale method is used to design and distribute a questionnaire to evaluate the satisfaction of the commercial space of Nanpu Times in terms of traffic flow and to obtain the satisfaction scores of each index to obtain the comprehensive evaluation results (Figure 4).

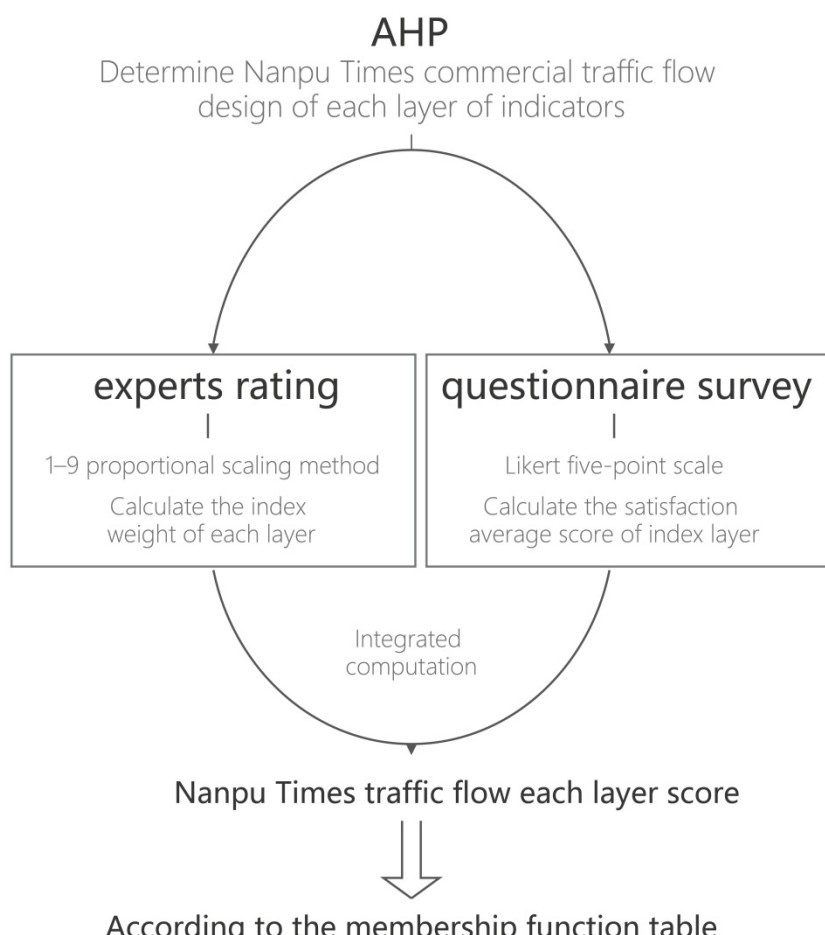

**Figure 4.** A schema of the methodology and application steps.

### 3.4. Traffic Flow Design and Construction of Satisfaction Evaluation Index System

The construction of the evaluation index system should follow the principles of systematisms, objectivity, scientificity, and practicability. The evaluation factors are divided according to the expert interviews, then the index weights are determined using social science data analysis methods, and finally, the satisfaction evaluation system is constructed [43]. Referring to the application principle of AHP, the satisfaction evaluation index system of Nanpu Times traffic flow is divided into four levels: target layer, criterion layer, factor layer, and index layer [37,38]. Through the desktop data collection method, it is decided that the basic principles of traffic flow design should be followed: comfort, accessibility, and identifiability [44,45]; the criterion layer is therefore defined as these three evaluation indexes. The traffic flow can generally be distinguished from indoor/outdoor

and horizontal/vertical flow [46]. Because Nanpu Times is a block commercial building space, this study adopts horizontal and vertical traffic flow as divisions. Based on this, the factor layer is divided into the horizontal and vertical traffic flow layers according to the three indexes of the criterion layer. We combined our professional knowledge of and the related literature (including the works of Gaoxiang Dong [47] and Huaidong Wu [48]) on the study of traffic flow design planning to determine the content in the index layer, and finally, through expert consultation, discussion, and research, we determined the Nanpu Times traffic flow satisfaction evaluation index system X (Figure 5).

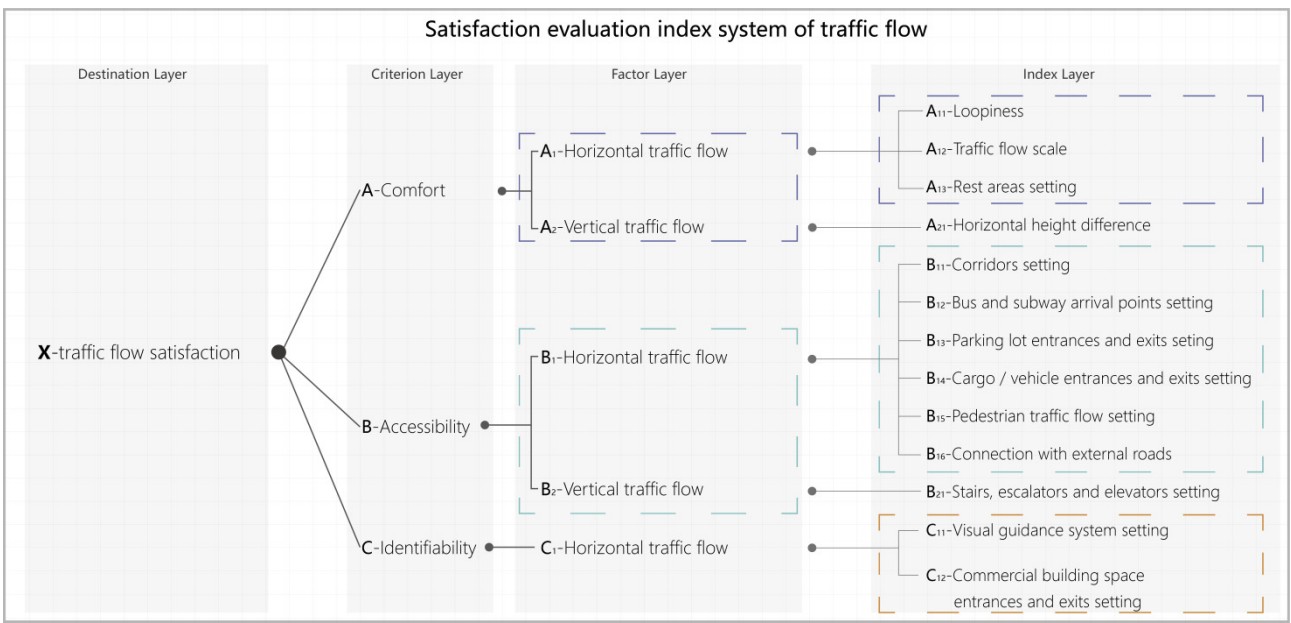

**Figure 5.** The satisfaction evaluation index system of Nanpu Times traffic flow design.

*3.5. Construction of Judgment Matrix and Weight Calculation of Satisfaction Evaluation Index of Traffic Flow Design*

Determining the standard and constructing the satisfaction evaluation index system does not necessitate the implementation of the final evaluation. We can create an effective, comprehensive evaluation system by calculating the weight value. This study uses AHP to analyze the data and selects seven experts in related fields. Three are female, and four are male. They work as PhD students, professors, and directors of interior and environmental design in the field of design studies. The experts were interviewed and allowed to judge the weights based on their own experience.

For the three-layer evaluation factor set constructed above, the 1–9 scale method (Table 1) was adopted for each element layer by layer to obtain the judgment matrix H.

$$H = \left(h_{ij}\right)_{n \times n} \tag{1}$$

where H is the comparison between the weights of two factors in a single layer, which meets the following criteria:

$$h_{ij} \geq 0, h_{ji} = \frac{1}{a_{ij}}, h_{ii} = 1 \tag{2}$$

$h_{ij}$ represents the number corresponding to row *i* and column *j*.

**Table 1.** 1–9 proportional scaling method.

| $a_{ij}$ Scale | Scale Meaning (Element *i* Is the Vertical Factor in the Table, Element *j* Is the Horizontal Axis Factor in the Table) |
|---|---|
| 1 | *i* has the same importance as *j* |
| 3 | Compared with the two factors of *i* and *j*, the horizontal is slightly more important than the vertical |
| 5 | Compared with the two factors of *i* and *j*, the horizontal is more important than the vertical |
| 7 | Compared with the two factors of *i* and *j*, the horizontal is much more important than the vertical |
| 9 | Compared with the two factors of *i* and *j*, the horizontal is absolutely more important than the vertical |
| 2, 4, 6, 8 | Intermediate value of *i* and *j* judgment |
| Reciprocal | If the importance ratio of factor *i* to factor *j* is $a_{ij}$, the importance ratio of factor *j* to factor *i* is $\frac{1}{a_{ij}}$ |

The indexes of seven experts are assigned, averaged, and rounded to obtain fair and objective results. We calculated the eigenvector $a_i$, which corresponds to the maximum eigenvalue $\lambda_{\max}$ of the judgment matrix H; that is, the weight assignment of the index factors.

1.  Normalization of each column of judgment matrix H:

$$P_{ij} = \frac{h_{ij}}{\sum_{i=1}^{n} h_{ij}}. \tag{3}$$

2.  Sum $P_{ij}$ by line:

$$P_i = \sum_{j=1}^{n} P_{ij}. \tag{4}$$

3.  Normalize $P_i$:

$$a_i = \frac{P_i}{\sum_{i=1}^{n} P_i}. \tag{5}$$

4.  Calculate the maximum characteristic heel of the judgment matrix:

$$\lambda_{\max} = \frac{1}{n} \sum_{i=1}^{n} \frac{(H_a)_i}{a_i}. \tag{6}$$

*3.6. Check the Consistency of Judgment Matrix*

1.  Check the consistency of the judgment matrix:

$$CR = \frac{CI}{RI} \tag{7}$$

where *CI* represents the consistency index of the judgment matrix and *RI* represents the random consistency index of the judgment matrix (Table 2).

**Table 2.** The random consistency index of the judgment matrix.

| Dimension | 1 | 2 | 3 | 4 | 5 | 6 | 7 | 8 |
|---|---|---|---|---|---|---|---|---|
| *RI* | 0 | 0 | 0.58 | 0.90 | 1.12 | 1.24 | 1.32 | 1.41 |

2.  Calculation of the consistency index of the judgment matrix:

$$CI = \frac{\lambda_{\max} - n}{n - 1} \tag{8}$$

where $n$ represents the dimension of the judgment matrix. When $CR < 0.1$, the consistency of the judgment matrix is acceptable.

### 3.7. Traffic Flow Design and Satisfaction Evaluation Survey Index Weight Determination

Using the AHP, taking the satisfaction evaluation index X of Nanpu Times traffic flow design as an example, calculate the index weight of its lower criterion layer index "Comfort A, Accessibility B, and Identifiability C". The calculation process is as follows:

1.  Obtain the average value of the importance score by the 1–9 proportional scale method from the questionnaire of seven experts.
2.  Calculate the index weight through the above formula:
    $a_1 = 0.161$, $a_2 = 0.765$, $a_3 = 0.074$ (Figure 6).
    The largest characteristic root is $\lambda_{\max} = \frac{1}{n} \sum_{i=1}^{n} \frac{(H_a)_i}{a_i}$, verifying the consistency of the expert scoring matrix, and we can draw the following conclusion: $\lambda_{\max} = 3.107$.
3.  Check the consistency of the judgment matrix:

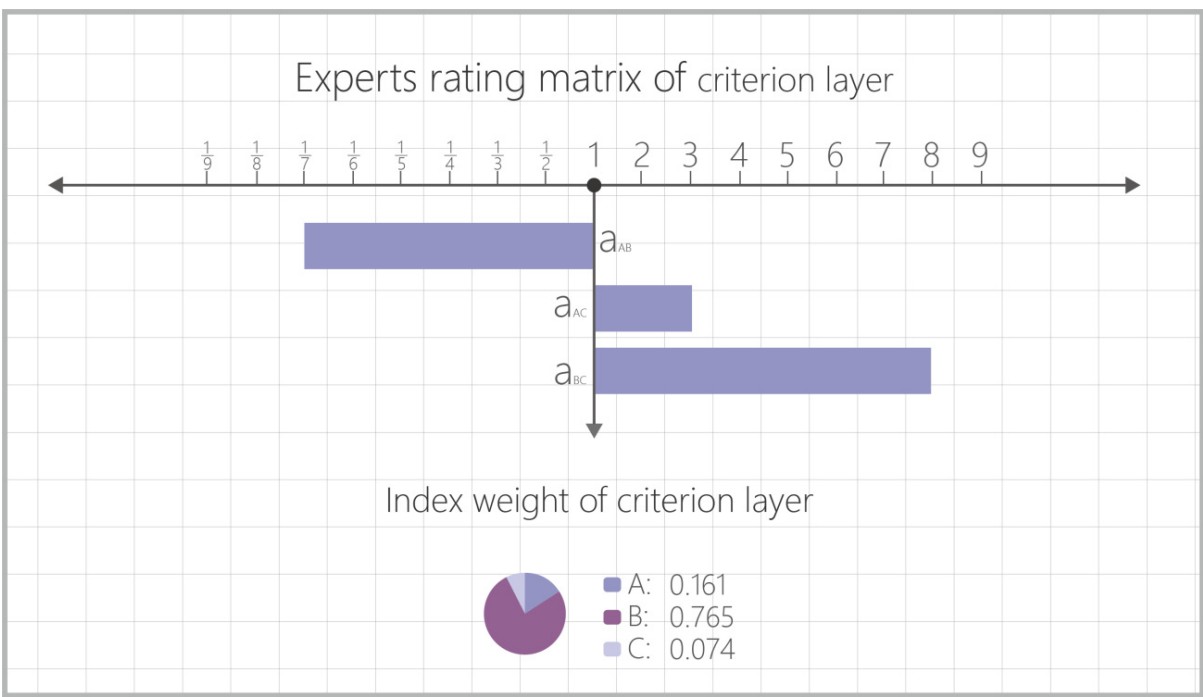

**Figure 6.** Experts' rating judgment and index weight of criterion layer.

$CR = \frac{CI}{RI}$, here $n = 3$ and $RI$ is 0.58. According to the above formula, $CI = 0.0535$, so $CR = 0.092 < 0.1$, meaning that the judgment matrix has satisfactory consistency.

The calculation results show that the experts consider Accessibility B (76.5%) to be the most important, followed by Comfort A (16.1%) and Identification C (7.4%) in Nanpu Times' traffic flow design.

We use the above steps to determine the weight of each factor of each layer (Figures 7 and 8).

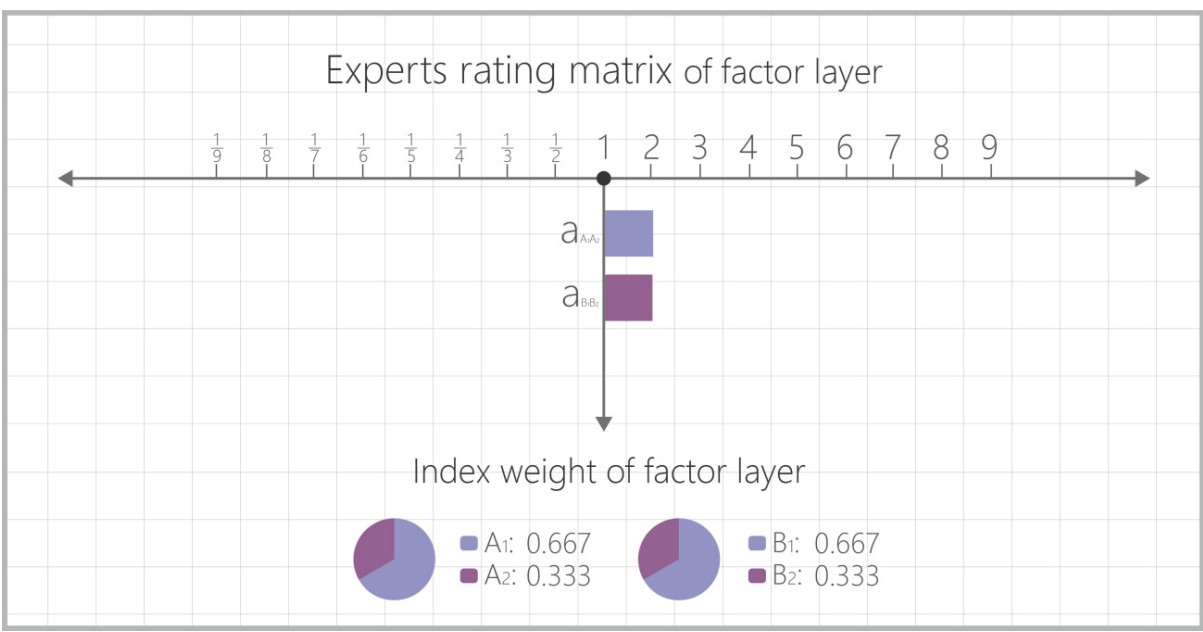

**Figure 7.** Experts' rating judgment and index weight of factor layer.

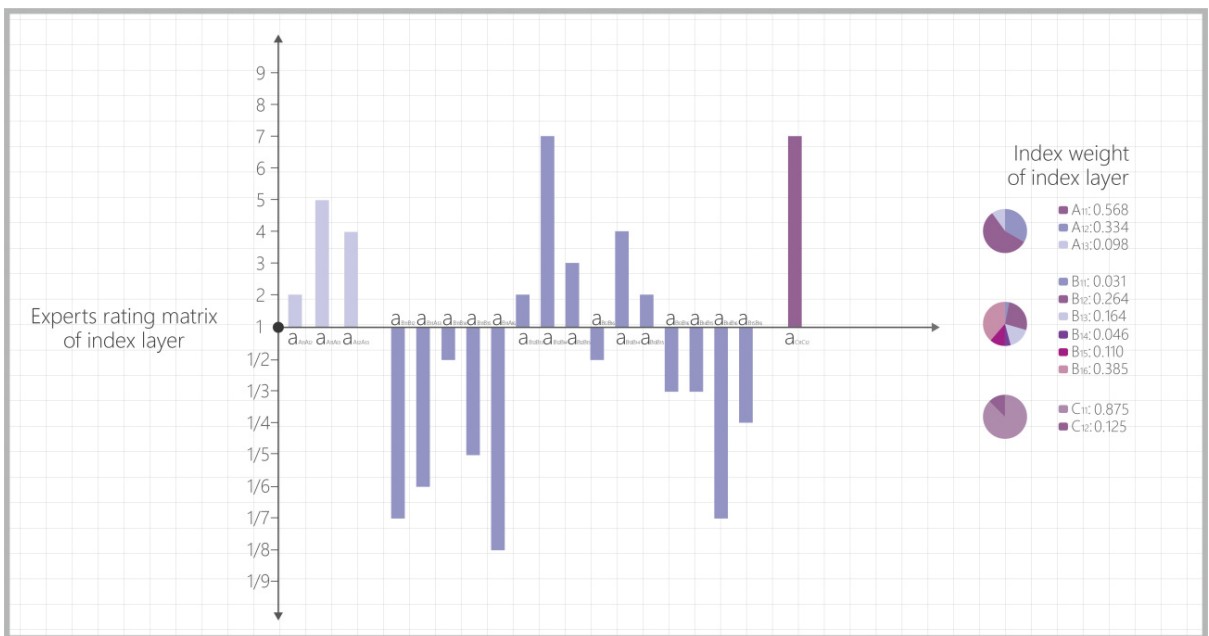

**Figure 8.** Experts' rating judgment and index weight of index layer.

The above data were calculated to obtain a weighting analysis of the factor set of the post-use satisfaction evaluation index system of the Nampu Times traffic flow. In the following stages, the satisfaction scores for the commercial space traffic flow design will be calculated based on these weights and then combined with the specific satisfaction scores obtained from the Likert scale questionnaire. Based on the support of the POE method, this provides a methodological practice for the subsequent design of such commercial space traffic flows. Figure 9 intuitively reflect the weight status of all evaluation factors (see Table A1 for data summary).

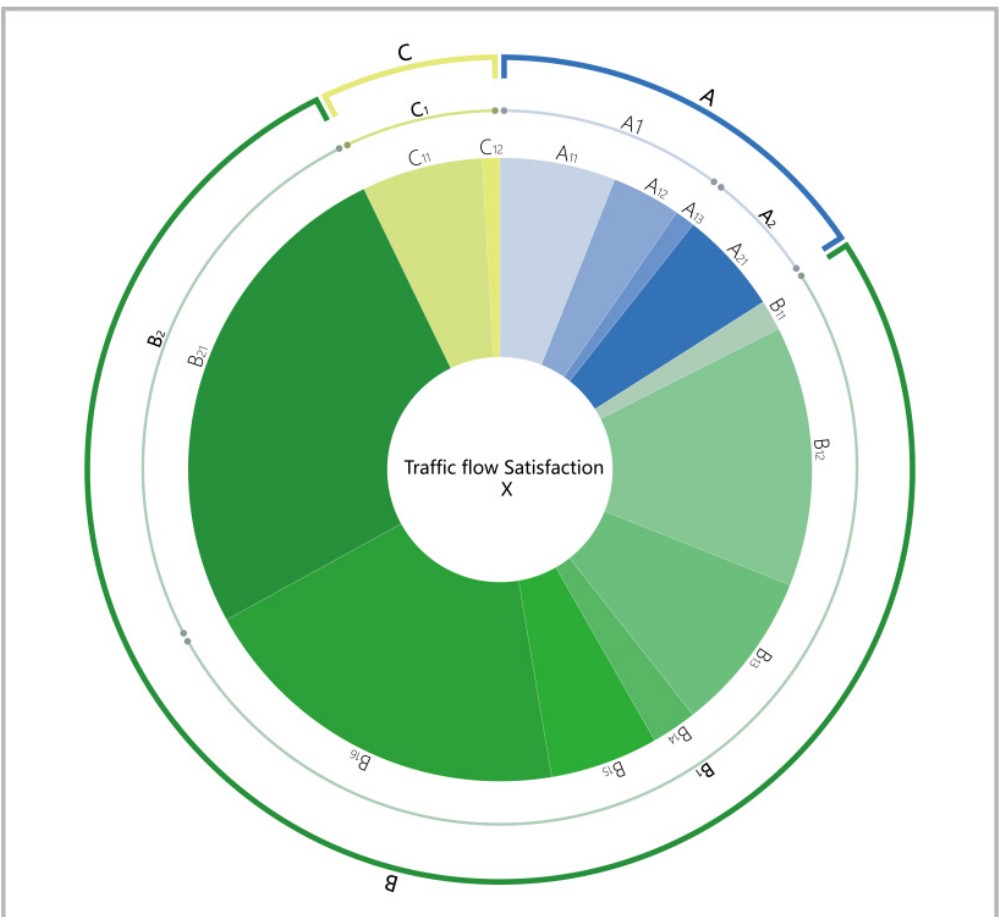

**Figure 9.** The index weight of the comprehensive evaluation index system.

### 3.8. Questionnaire Design

After constructing the satisfaction evaluation index system and obtaining the weight values of each index, the satisfaction score of each index needs to be accepted, which is calculated using a questionnaire in this study. The questionnaire is divided into two parts: the first part collects the basic information of the respondents, including gender, age, address, and other essential information. The second part is the respondents' evaluation of their satisfaction with the Nanpu Times traffic flow design. It uses the five-point Likert scale method, with the degree of psychological response to each indicator being divided into five levels: very dissatisfied, not very satisfied, average, relatively satisfied, and very satisfied, with values of 1, 2, 3, 4, and 5 being assigned to them, respectively.

### 3.9. Questionnaire Distribution

The questionnaires were distributed online and offline simultaneously from 1–15 January 2022. The offline questionnaires were distributed between 9:00–10:00 and 18:00–19:00, during the peak commuting hours and mealtimes, when consumers were randomly selected and distributed. The online questionnaire was distributed to Nampu Times staff and surrounding community owners to ensure the authenticity and validity of the data and to guarantee the sample size of the questionnaire. We distributed a total of 206 online questionnaires, of which 199 were useful, and 100 offline questionnaires, of which 95 were useful, making for a total of 293 useful questionnaires being collected.

In this study, the reliability of the sample data was tested through the Cronbach $\alpha$ coefficient using SPSS 24.0 software (SPSS Inc., Chicago, IL, USA), and the results showed a reliability coefficient value of 0.949, which is greater than 0.9, indicating that the research data are of high reliability and can be used for further analysis. Then, the validity analysis

was carried out through SPSS, and the results showed that the KMO value was 0.915 and the significance was 0.000, indicating that the validity of the questionnaire was good and that the research items were reasonable and meaningful.

## 4. Results and Discussion

### 4.1. Satisfaction Scores for Nanpu Times Traffic Flow Design

Among the 293 respondents, there were 170 females, accounting for 58.02%, and 123 males, accounting for 41.98%. Among them, people aged 18–25 represented the biggest group, accounting for 44.71%, followed by those aged 26–40, accounting for 37.2%. It can be seen that for commercial building spaces, young and middle-aged women comprise the main group. Most respondents came from within Guangzhou and the neighborhood, accounting for 51.54% and 35.84% of respondents, respectively. A total of 51.19% of respondents were with their classmates, colleagues, and friends, while 19.8% were with their partners. Most respondents traveled by bus/subway or on foot, accounting for 48.46% and 23.55% of respondents, respectively. When the respondents came to Nanpu Times, their main activities were socializing, consumption, resting, and entertainment (Figure 10). A separate questionnaire on the cargo/vehicle entrance and exit settings was collected from the staff and was credible, with 90 responses. A large proportion of the respondents were female and mainly from the 18–25 age group within Guangzhou, so they are more likely to travel by bus and subway, which may make them more concerned about the location of bus and subway arrival points.

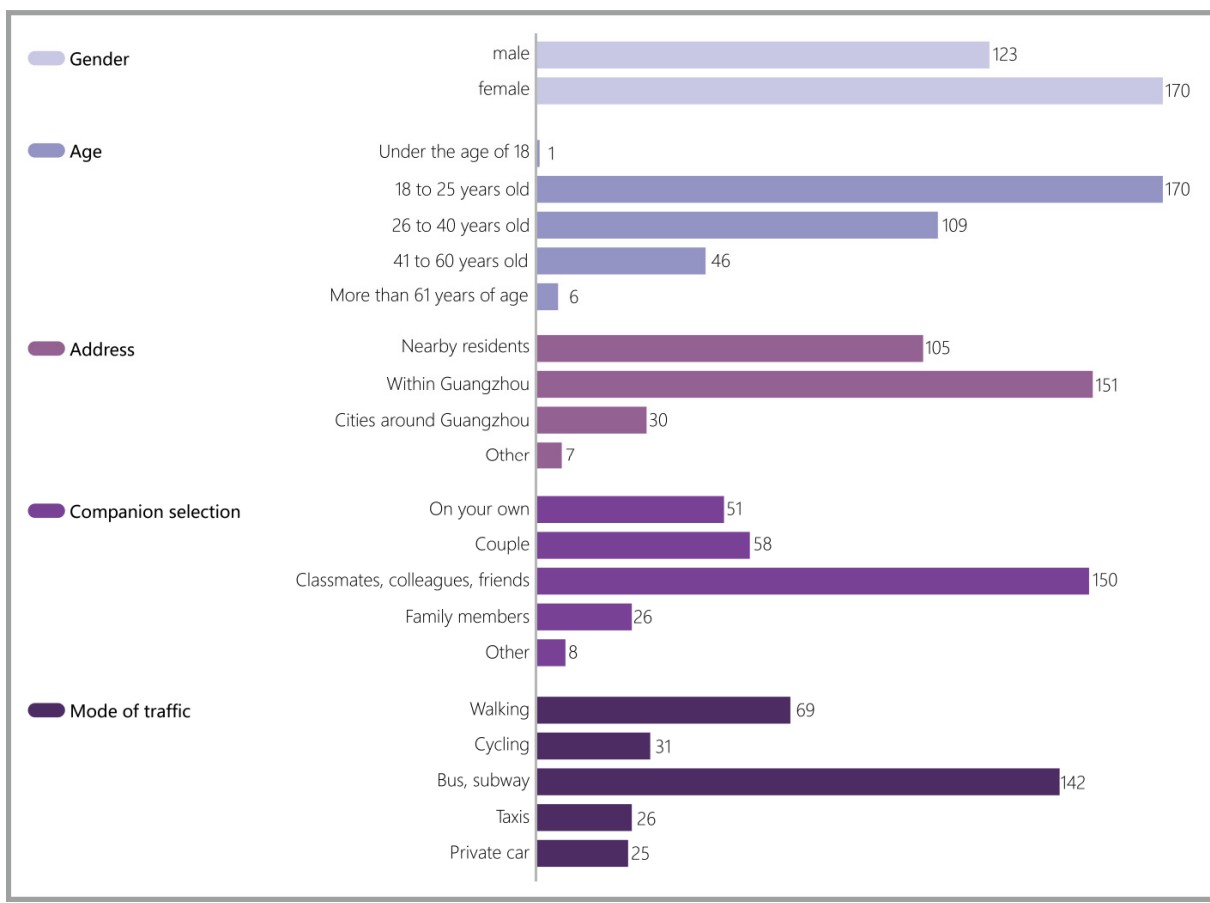

**Figure 10.** Information on residents interviewed.

Through questionnaires and statistics, the satisfaction score results of each index were obtained (Figure 11) (see Table A2 for data summary). It can be concluded from Figure 11 that the three indexes with the highest scores are the evaluation of bus and subway arrival points (4.11), the evaluation of roundness (3.32), and the evaluation of traffic flow scale (3.27). The five indexes with the lowest scores are the evaluation of visual guidance system settings (2.36), stairs, escalators, and elevator settings (2.41), parking lot entrance and exit settings (2.44), commercial building space entrance and exit settings (2.55), and cargo/vehicle entrance and exit settings (2.7), indicating that the respondents have the lowest satisfaction evaluation for these aspects. We put forward optimization suggestions for these five items specified below.

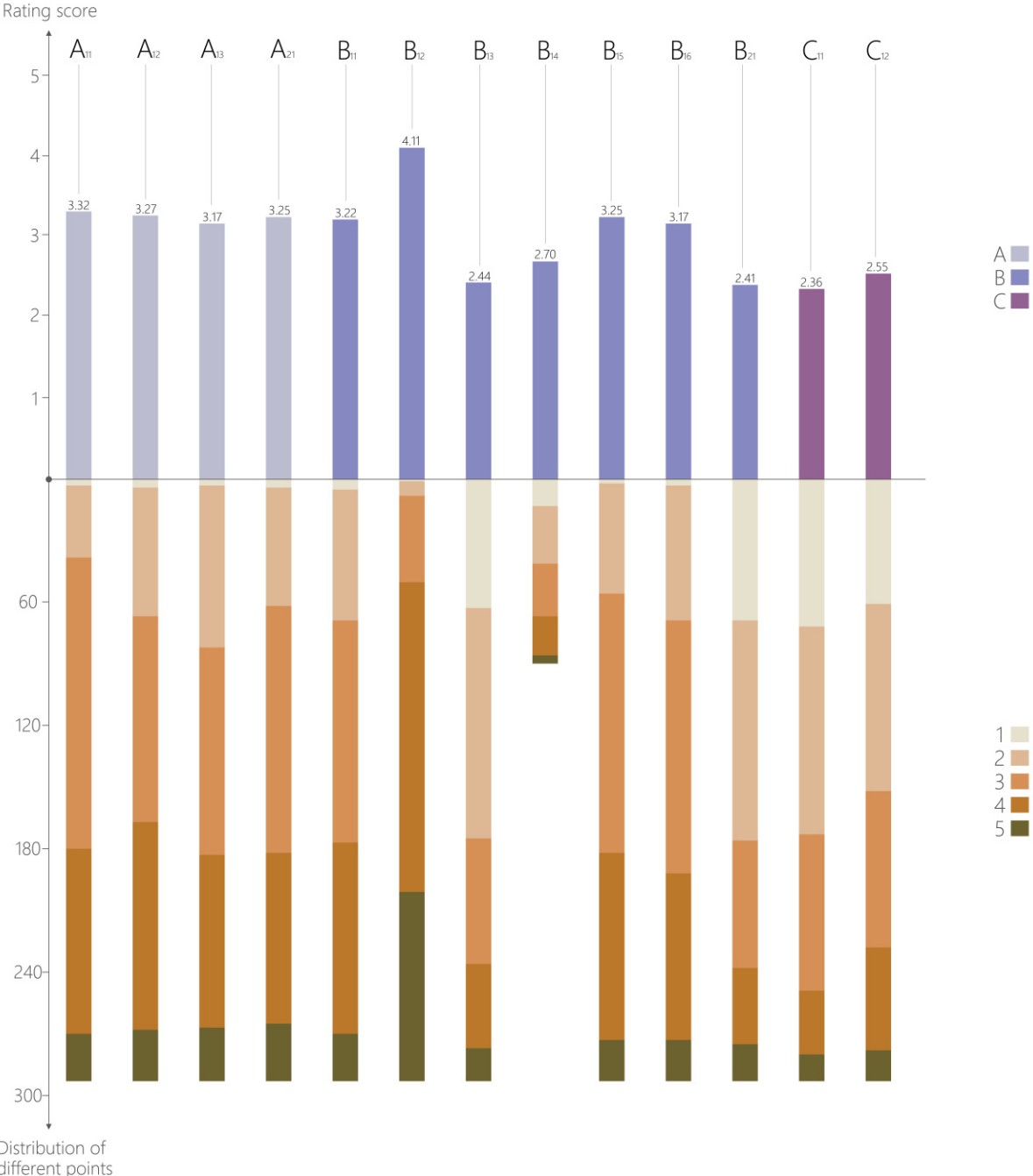

**Figure 11.** Statistical table of questionnaire results for the Nanpu Times traffic flow design.

According to the score of the index layer and the weight distribution, the scores for the factor layer, the criterion layer, and the target layer were calculated (Table 3).

**Table 3.** The scores for the factor layer, criterion layer, and destination layer.

| Destination Layer | Score | Criterion Layer | Index Weight | Score | Factor Layer | Index Weight | Score |
|---|---|---|---|---|---|---|---|
| X | 2.995 | A | 0.161 | 3.276 | $A_1$ | 0.667 | 3.289 |
| | | | | | $A_2$ | 0.333 | 3.250 |
| | | B | 0.765 | 2.995 | $B_1$ | 0.667 | 3.287 |
| | | | | | $B_2$ | 0.333 | 2.410 |
| | | C | 0.074 | 2.384 | $C_1$ | 1.000 | 2.384 |

According to the membership function table (Table 4), it can be concluded that the satisfaction degree of Nanpu Times traffic flow design is rated E3; general comfort and accessibility are also rated E3; and the identifiability score is the lowest, which is E2.

**Table 4.** Membership function table.

| Evaluation of Estimate Z | Common | Rate |
|---|---|---|
| $Z \leq 1.5$ | Very dissatisfied | E1 |
| $1.5 < Z \leq 2.5$ | Dissatisfied | E2 |
| $2.5 < Z \leq 3.5$ | General | E3 |
| $3.5 < Z \leq 4.5$ | Quite satisfied | E4 |
| $Z > 4.5$ | Very satisfied | E5 |

*4.2. Strategy for Commercial Building Space Traffic Flow Design*

This study calculated the weights of the comfort, accessibility, and identifiability of the three principles in the criterion layer, and it found that the accessibility weight score was the highest, the comfort weight score was between the accessibility and identifiability, and the identifiability weight score was slightly lower than the comfort and accessibility. Based on the satisfaction evaluation results of Nanpu Times traffic flow design, this paper puts forward planning suggestions for the traffic flow design of the same type of commercial building space in the future according to the three principles of the criterion layer. According to the principles, we analyze the five indexes with the lowest satisfaction evaluation of Nanpu Times: the evaluation of the visual guidance system settings (2.36), the evaluation of stairs, escalators, and elevators settings (2.41), the evaluation of the parking lot entrances and exits settings (2.44), the evaluation of commercial building space entrances and exits settings (2.55), and the evaluation of cargo/vehicle entrances and exits settings (2.7), and put forward optimization suggestions.

4.2.1. Design Suggestions Are Put Forward According to the Comfort

For each index factor in comfort, firstly, we should consider the traffic flow loopiness and scale of commercial building space. The higher the traffic flow loopiness, the better the traffic flow in theory, so users do not go back. Too much secondary traffic flow will affect the return degree. However, provided that the guidance is reasonable, the secondary traffic flow can also improve the shopping experience. This design also requires us to consider the identifiability. Secondly, the rest area settings also play a crucial role in commercial building space. They should meet users' psychological and physiological needs and design a humanized and characteristic rest area in combination with the characteristics of commercial building space. It can also be combined with the surrounding landscape facilities, which not only reduces the monotonicity of the rest area but also meets the needs of users. In addition, the rich horizontal height difference can increase the sense of traffic flow hierarchy of commercial building space and give vitality to the whole traffic flow.

4.2.2. Design Suggestions Are Put Forward for the Principle of Accessibility

As accessibility has the highest weight score, each index is critical. For each index factor in accessibility, we first deal with the relationship between people flow and vehicle flow in horizontal traffic flow. Based on the people-oriented principle [8,49], the traditional mixed mode of people and vehicles is no longer applicable to modern commercial building space. Faster moving vehicles are more likely to cause accidental injury to pedestrians. They are also more likely to cause traffic jams resulting in congestion, which does not facilitate consumer movement and reduces the desire to buy. The separation of people and vehicles can meet the requirements of vehicle convenience and pedestrian safety and form a convenient and accessible traffic flow environment. The pedestrian traffic flow and the commercial format can be considered in the pedestrian route, which leads the user step by step into the interior of the commercial space through changes in color and floor tiles, creating a visual continuity. Secondly, the number and location of stairs, elevators, and escalators shall be set up in the vertical traffic flow, which connects the vertical direction of commercial building space. In design, each node of commercial building space can be used reasonably and effectively, and escalators are used primarily to increase the positive polarity of users flowing to the upper space, forming a crisscross three-dimensional flow system. Through post-use index evaluation and user interviews, our specific improvement suggestions for each index of Nanpu Times are as follows:

1. The cargo/vehicle entrance and exit settings need to be improved (Figure 12). Firstly, the road is too narrow, and the size of large trucks needs to be considered. Secondly, it is necessary to plan the access lane route to reduce the cross-traffic between access vehicles and congestion.

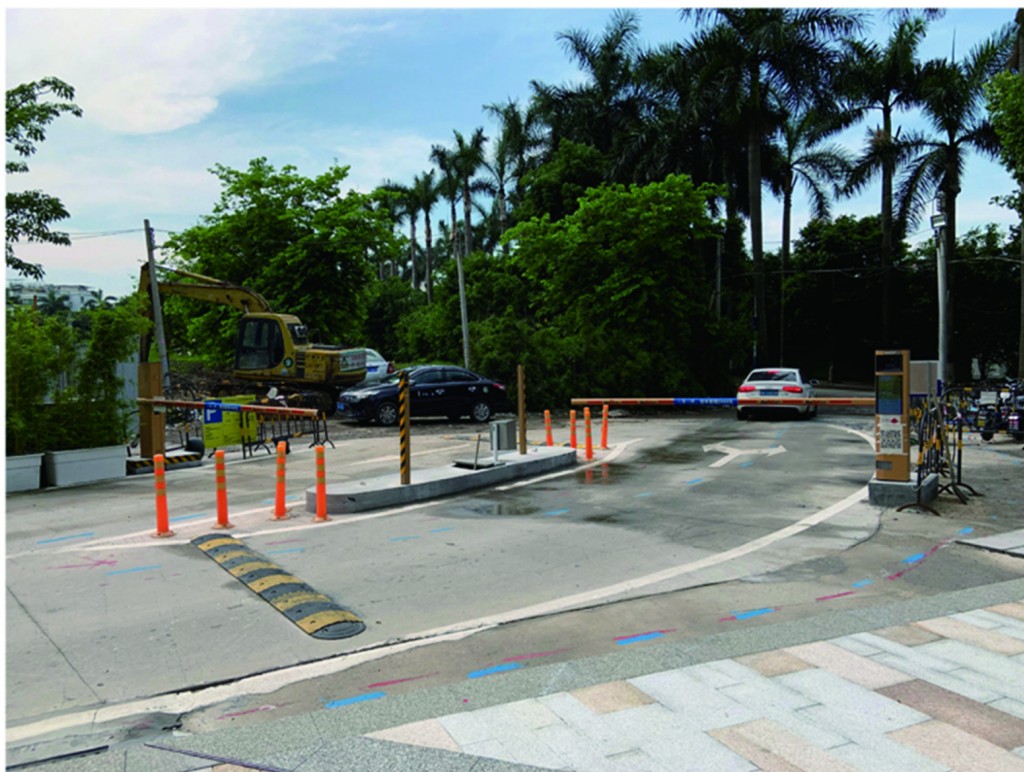

**Figure 12.** Current situation of cargo/vehicle entrances and exits of Nanpu Times.

2. The parking lot entrance and exit settings need to be improved. Figures 12 and 13 show the current situation of the car park entrances to the commercial space. The above shows that most of the users arrive at Nanpu Times by way of bus/subway, and the parking lot entrances and exits of this commercial space are designed to affect each other by the design of the traffic entrance and entrance line on top of

the pedestrian line after the subway arrival point. The pedestrian and vehicle flows should be separated and not interfere with each other.

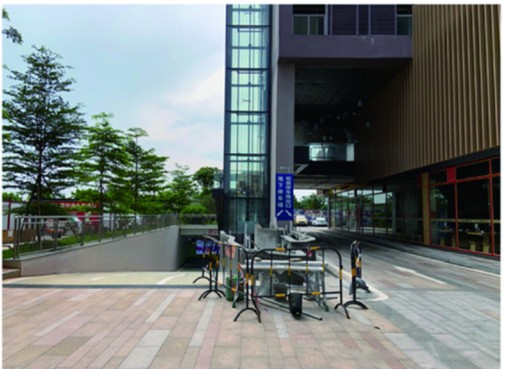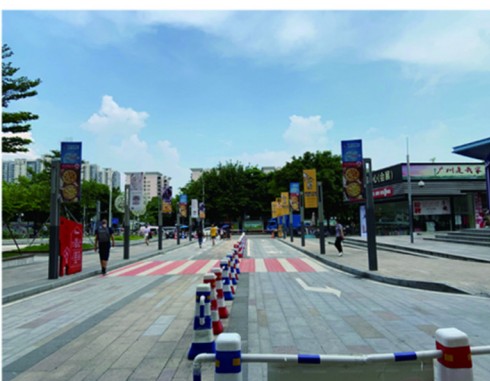

**Figure 13.** Current situation of stairs, escalators, and elevators of Nanpu Times.

3.    The stairs, escalator, and elevator settings need to be improved (Figure 14). These settings are too isolated and lack interactive elements, resulting in a lack of consumer interaction and participation. These problems can be solved by adding interactive devices or combining the design of landscape nodes to form a coherent three-dimensional traffic flow.

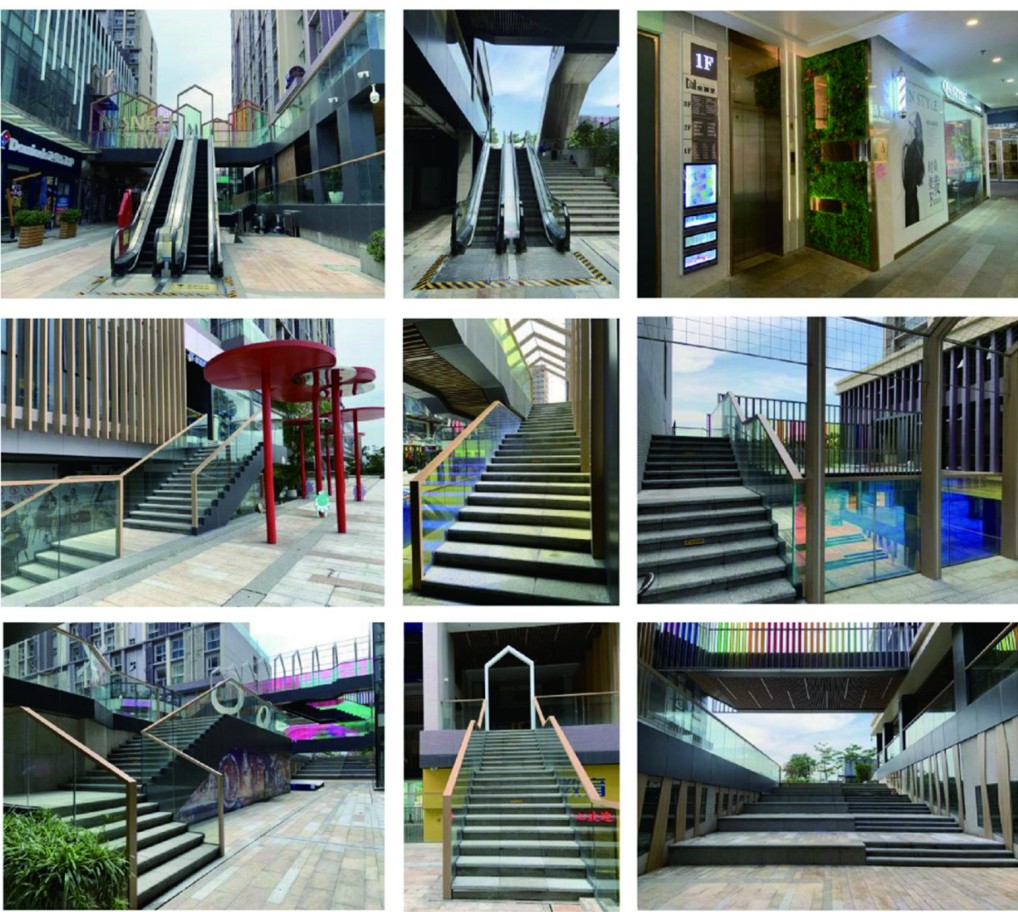

**Figure 14.** Current situation of parking lot entrances and exits of Nanpu Times.

### 4.2.3. Design Suggestions Are Put Forward for the Principle of Identifiability

The first step is to tackle visual guidance system settings for each index factor in identifiability. A visual orientation system is an effective pictorial or textual indication that guides the user to information and allows the user to perceive the space quickly and effectively. An excellent visual guidance system can effectively improve the spatial image and service quality. Secondly, the commercial building space entrances and exits are a bridge between the commercial building space and the outside; these can be designed in combination with the nature of the business. Nanpu Times' traffic flow design recognition score is E2, which significantly reduces the overall traffic flow design satisfaction evaluation score. The specific improvement suggestions for the indicators of Nanpu Times are as follows:

1. The visual guidance system settings need to be improved (Figure 15). Firstly, they should be more concise and clear. The design should be carried out in the form of visualization, marking the traffic flow of the entire commercial building space in the form of a commercial building space plan, including arrow indicators marking their location. Secondly, there are too few visual guidance system settings for traffic flow in commercial building space, merely at the top and tail, which should be increased appropriately. Thirdly, the settings in the squares outside the commercial building space are numerous but not sufficiently clear. There are quite a few instructions, and the result of extending in all directions may be a circle, which can lead to users not having clear indicators and greatly affecting the identifiability of traffic.

2. The commercial building space entrance and exit settings need to be improved (Figure 16). They are an important factor in introducing external people into the interior. They should be designed to be different from other buildings' entrances and exits and to highlight the characteristics of the theme of Nanpu Times commercial space, with a more personalized entrance that can increase recognition.

In the next step of the study, the team will work with the Nanpu Times commercial space operators to obtain information on the change in traffic flow through machine measurements. Specific information on pedestrian and vehicular flows will be obtained, and we aim to measure whether the critical nodes of the traffic flow have been improved. We will verify the effectiveness of the research strategy in multiple dimensions and iterate the traffic flow design based on the results of our analysis.

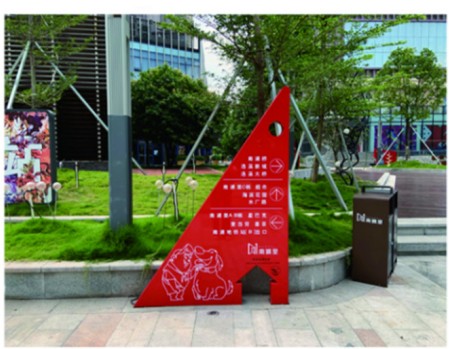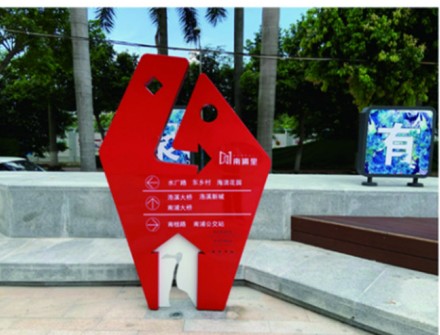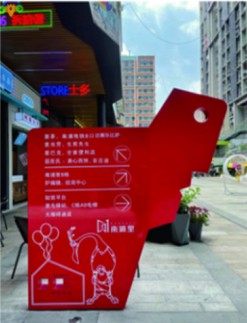

**Figure 15.** Current situation of the visual guidance system of Nanpu Times.

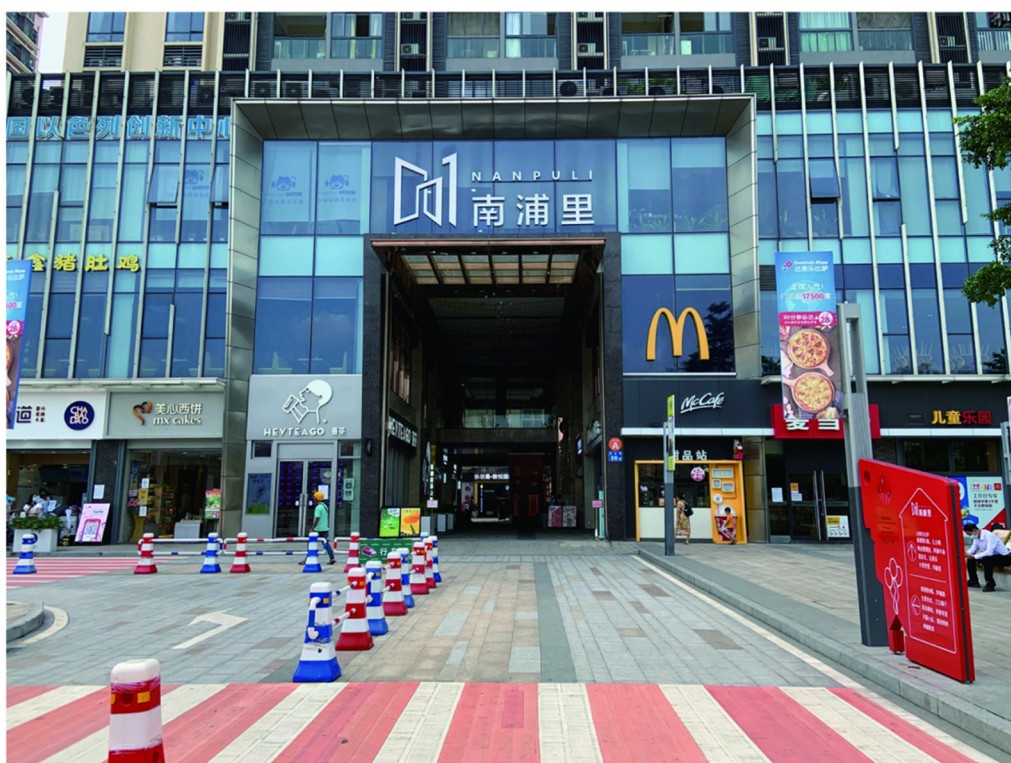

**Figure 16.** Current situation of the entrances and exits of Nanpu Times.

## 5. Conclusions

This study was based on POE and used AHP to construct a traffic flow evaluation index system for commercial building space. Then, expert interviews and questionnaires were used to understand users' satisfaction with the indexes after experiencing the facilities and we calculated each index's weight and satisfaction score. Finally, we analyzed the evaluations and causes of the traffic flow design of commercial building space to obtain the key points. The overall satisfaction with the commercial building space traffic flow design of Nanpu Times in general and three design strategies for comfort, accessibility, and identifiability were proposed to solve the existing traffic flow problems. This study found that the construction of the satisfaction evaluation index system, the weighting analysis method, the questionnaire survey method, and some numerical statistical analysis, such as the introduction of descriptive and inferential statistics, can be applied to the same type of research using POE when studying the traffic flow design in commercial building spaces, but there are still some limitations. This investigation is only a case study of the Nanpu Times commercial building space, which is not universal, and it is not known whether it can be fully applied to the traffic flow design in all commercial building spaces, such as office buildings. In future research, we could use POE more to study the traffic flow design to make it universally applicable. Although this evaluation method works well in other fields, there are some specificities in the study of commercial space traffic flow design, such as consumer psychological needs, behavioral changes, etc. By understanding the needs and wants of users, we can evaluate, analyze, and summarize the problems existing in its traffic flow design and make reasonable suggestions for optimization. Eventually, we can provide a scientific, reasonable, and practical basis for the subsequent traffic flow design of the same type of commercial building space to drive the prosperity and development of commercial building space.

**Author Contributions:** Conceptualization, Y.Z. and L.H.; methodology, Y.Z.; software, Y.Z.; validation, Y.Z., L.H. and W.H.; formal analysis, L.H.; investigation, Y.Z. and W.H.; resources, L.H.; data curation, Y.Z.; writing—original draft preparation, Y.Z.; writing—review and editing, Y.Z.; visualization, W.H.; supervision, L.H.; project administration, L.H.; funding acquisition, L.H. All authors have read and agreed to the published version of the manuscript.

**Funding:** This research was funded by the 14th Five-Year Plan for the Development of Philosophy and Social Science in Guangzhou province, grant number 2021GZGJ283.

**Institutional Review Board Statement:** Approved by Specialized Committee on Academic Ethics and Ethics on Science and Technology of the Guangdong University of Technology.

**Informed Consent Statement:** Informed consent was obtained from all subjects involved in the study.

**Data Availability Statement:** Not applicable.

**Conflicts of Interest:** The authors declare no conflict of interest.

## Appendix A

**Table A1.** Data summary form of the index weight of comprehensive evaluation index system.

| Destination Layer | Criterion Layer | Index Weight | Factor Layer | Index Weight | Index Layer | Index Weight |
|---|---|---|---|---|---|---|
| | A | 0.161 | $A_1$ | 0.667 | $A_{11}$ | 0.568 |
| | | | | | $A_{12}$ | 0.334 |
| | | | | | $A_{13}$ | 0.098 |
| | | | $A_2$ | 0.333 | $A_{21}$ | 1.000 |
| Guangzhou Nanpu Times commercial space traffic flow design satisfaction evaluation index system X | | | | | $B_{11}$ | 0.031 |
| | | | | | $B_{12}$ | 0.264 |
| | B | 0.765 | $B_1$ | 0.667 | $B_{13}$ | 0.164 |
| | | | | | $B_{14}$ | 0.046 |
| | | | | | $B_{15}$ | 0.110 |
| | | | | | $B_{16}$ | 0.385 |
| | | | $B_2$ | 0.333 | $B_{21}$ | 1.000 |
| | C | 0.074 | $C_1$ | 1.000 | $C_{11}$ | 0.875 |
| | | | | | $C_{12}$ | 0.125 |

**Table A2.** Data summary form of statistical table of questionnaire results.

| Index Factors | Score | | | | | The Average |
|---|---|---|---|---|---|---|
| | 1 | 2 | 3 | 4 | 5 | |
| $A_{11}$ | 3 | 35 | 142 | 90 | 23 | 3.32 |
| $A_{12}$ | 4 | 63 | 100 | 101 | 25 | 3.27 |
| $A_{12}$ | 3 | 79 | 101 | 84 | 26 | 3.17 |
| $A_{21}$ | 4 | 58 | 120 | 83 | 28 | 3.25 |
| $B_{11}$ | 5 | 64 | 108 | 93 | 23 | 3.22 |
| $B_{12}$ | 1 | 7 | 42 | 151 | 92 | 4.11 |
| $B_{13}$ | 63 | 112 | 61 | 41 | 16 | 2.44 |
| $B_{14}$ | 13 | 28 | 26 | 19 | 4 | 2.7 |
| $B_{15}$ | 2 | 54 | 126 | 91 | 20 | 3.25 |
| $B_{16}$ | 3 | 66 | 123 | 81 | 20 | 3.17 |
| $B_{21}$ | 69 | 107 | 62 | 37 | 18 | 2.41 |
| $C_{11}$ | 72 | 101 | 76 | 31 | 13 | 2.36 |
| $C_{12}$ | 61 | 91 | 76 | 50 | 15 | 2.55 |

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
