# Peer review of "Research on a Commercial Building Space Traffic Flow Design Based on Post-Occupancy Evaluation"

_buildings, doi:10.3390/buildings12060838_

Round 1
Reviewer 1 Report
The resubmitted version of this paper exhibits major improvements. Few notes for consideration here;
· The suggested title on the response form fits better the scope of the paper, “Commercial Building Space Traffic Flow Design Based on the Post-Occupancy Evaluation” maybe consider adding a case study given that it is clearly stated in the conclusion that the results are specific and may not be generalized.
Note that however the uploaded manuscript still shows the previous submission title. Recommend updating it.
· Clarification of the concept, variables, and methodology have been well-enhanced. Minor text editing.
Reviewer 2 Report
I appreciated your efforts as most comments and suggestions have been implemented in the revised manuscript. The introduction and literature review sections have in fact been improved in the revised version of the manuscript. In addition, the schema added in the methodology section, however simple, can help to better understand the steps of methodology and application, and the revisions made throughout the manuscript make it easier to read. Finally, the English has been improved, but it could be further improved in the final version.
Reviewer 3 Report
Overall, the work has value and will be interesting to potential readers in the AEC industry. However, it needs significant revision especially with grammar, sentence structure, and the format of the paper. I initially was commenting on changes for the grammar and sentence restructuring but as there were significant issues I stopped. I recommend reviewing the whole paper including the figures and Tables and making changes as needed.
The full comments are in the word document and also included below.
Lines 9 – 10: Suggest updating first sentence in abstract for easier reading. See suggestion “The aim of traffic flow design in commercial building space planning is to create a comfortable, easily accessible and identifiable spatial structure in a complex and changeable business environment”
Line 14: The abstract mentions that this study selects the commercial project "Guangzhou Nanpu Times Commercial Building Space" planned and operated by the team. Does the team include authors of this paper? If so, you should make it clear or describe the team that it was planned and operated by.
Lines 18 – 20: The sentences are incomplete and as a result are not clear on the message. Consider rephrasing with complete sentences and consider replacing ”So as to” with “To” or “Consequently” or “In order to”.
Line 26: Include a proper Segway between the first sentence in line 25 and this sentence on the development direction of commercial building space. It currently seems disjointed and does not provide ample background prior to this sentence.
Line 28: The sentence is incomplete. You may either combine with the previous sentence using a semi-colon or state that the development direction of commercial building space also brings challenges to the design of commercial building space traffic flow.
Line 29 – 31: The sentence is not clear, please consider rewriting.
Lines 31 – 33: You have not defined which commercial buildings are being addressed in this paper, so focusing only on purchasing behavior is a bit premature. What if it is an office building performing general business operations that do not require commerce, does the issue of traffic flow design also apply to it?
Lines 43 – 45: The sentence could be re-written
Lines 45 – 47: Re-write the sentence, it is not too clear.
Lines 49 – 52: Re-write the sentence
Line 93: Post occupancy evaluation (POE) does not include the use of scientific instruments. It only includes questionnaires/surveys and expert interviews. However, post occupancy evaluation and measurements (POE+M) a method established by the team at the Center for Building Performance (CBPD) using the NEAT cart uses instruments. You should consider restating the claim from POE to POE+M and citing work by the CBPD.
Line 110: Is the statement that only a few scholars have focused on commercial building space contrasting the earlier write-up in line 107 where research studies for POE in offices is mentioned?
Lines 168 – 174: Re-write the sentence, it is not clear and difficult to understand
Lines 165 – 167: You stated, “Through expert interviews and using AHP in conjunction with its own experience, it determines the individual evaluation indicators of the kinetic design, and through matrix scoring and quantitative analysis, determines the weight values of each index.” Does this mean the AHP weighting of the criteria was conducted by the authors?
Lines 184 - 187: “The Lichter scale method is used to design and distribute the questionnaire for evaluating the satisfaction of the commercial space of Nanpu Times in terms of movement lines, and to obtain the satisfaction scores of each index to obtain the comprehensive evaluation results (Figure 4).”
- Explain which parts of the Lichter scale methods were used to design and distribute the questionnaire. If the full litcher scale method was utilized, explain how it was used as a research method in this study
- Additionally, describe the profile of the respondents of the POE questionnaire and the recruitment method utilized, and rationale for the recruitment
Lines 195 – 196: “Referring to the application principle of AHP, the satisfaction evaluation index system of Nanpu Times traffic flow is divided into four levels: target layer, criterion layer, factor layer, and index layer”
- Are you adapting the AHP method? If you are adapting it/modifying please say so, because the AHP by Saaty uses the levels of goal/objective, main criteria, sub-criteria, sub-sub criteria, n level criteria, and alternatives as the final level. Your levels of target layer, criterion layer, etc., are confusing. Also, you may define how your own four levels relate to the main AHP levels by Saaty
- Also, cite the application principle of AHP that is being referred to, or define it
Line 220: “For the three-layer evaluation factor set constructed above, the 1-9 scale method (Table 1) was adopted for each element layer by layer to obtain the judgment matrix H.”
- State that it is the Saaty scale of pairwise comparison and cite.
Note: There needs to be significant overhaul of the grammar even within figures. For example, figures 6, 7 and 8 should be weight not “weigh”.
Lines 273 - 281 could be rewritten to be more succinct and clearer. For example, why did you call it POE methodology theory rather than just say POE methodology?
Lines 318 - 326 are very confusing. You make reference to figure 10 which illustrates respondents’ profile and responses for transportation traffic modes. However, I am unsure where the other indexes including “evaluation of roundness”, and traffic flow index came from in figure 10? If it is somewhere else, please reference it.
Line 363: Is it “each index inaccessibility is particularly important” or is it supposed to be each index in accessibility?
Line 365: “Based on the principle of being people-oriented” what principle? From where? Please cite. Stating principles and theories without references or definition is a common mistake in this paper. Also, explain how from the Principle of being people-oriented, the traditional mixed mode of people and vehicles is no longer applicable to modern commercial building space. Also elaborate on the relationship with the inaccessibility index.
Lines 441 - 442 counters your claim that POE can be applied to the commercial building space traffic flow design. It casts doubts on the research study and results of the study. I would recommend explaining the types of commercial building designs that your research study may not apply and/or the types it can be. Right now, you just cast a wide net that it may not apply to all commercial building spaces and didn’t explain which types and rationale for why it won’t apply, beyond it being a case study. Case studies can be transferable, and a few may be generalizable. What aspects of your case study can be transferable and which aspects can be generalizable? That would be good in the methodology. A deeper analysis of your case study. I would recommend reading Groat and Wang (2002) titled “Architectural Research Methods” chapter on case study to help.
Additionally, you state “Secondly, the total number of questionnaires is not very sufficient.” Why is it not sufficient and how does the limited number influence your results and the case you’re building? Just saying it is not sufficient is not building confidence in your research. What is your metric you’re using for sample size and how does the total of both your paper and online survey not meet the ideal sample size? Is there anything about the sample that still reflects the population from where you drew this sample from? Fo4 example, the population that utilizes commercial spaces is primarily 18–50-year-olds with xxx professions and sex as is reflected by this sample.
Results
I would also suggest rewriting your results section. Have a section for results/findings, and another section for discussion. Most if the write up currently in the results section is a discussion.
Put your findings on the POE questionnaire from respondents in the results. Also put the findings if the AHP analysis in the results. You should also consider including in the discussion how the sociodemographic characteristics of the POE respondents influences the evaluation of factors such as accessibility. Lastly, did you mention the sociodemographic of the experts who helped weight the criteria? If you did, please make it clearer if you did not, please include it.
Include future work/next steps of the research study in the discussion section.
Your conclusion should be stronger that its current form. Reiterate the problem and case, briefly mention the methods used and how it addresses the gaps, briefly mention the findings, and the implications/outcomes of the study.
Discussing bias: it may be beneficial to discuss the authors biases because the commercial building analyzed in this study is developed by the authors. How was the bias mitigated/minimized in this study?
Ethics: was IRB approval gotten for the human subject’s aspect of the research? Or permission sought in some other way? If so, please explain because it is not stated in the paper.
Methodology: Additionally, the mixed methods of POE and AHP is currently not clear and would need significant clarification on how they worked together - I have an understanding but it is not immediately clear. For example, what was in the POE questionnaire - show an example of the questions or include the full questionnaire as a supplemental. How did the questions inform the AHP criteria selection? It is written but not easy to understand.

Round 2
Reviewer 3 Report
Thank you for addressing my initial comments. The work improved significantly, however, it still requires revisions focused on proof reading, proper numbering, and grammar. I did not comment on minor grammatical errors but please take some time to proof-read the entire paper and make corrections as needed.
Please find some comments for revisions.
Lines 11 - 12: The sentence should be removed as it repeats the prior sentence in lines 9 - 10.
Line 16: ".as the subject." is a sentence fragment, please rewrite the whole sentence correctly. You also do not explain if the team are the authors or who the team is (as you stated you did in the cover letter response).
Lines 20 - 23: The two sentences are fragmented and unclear. Please re-write! I would once again recommend getting an editor to help proof-read and fix the errors in grammar, sentence structure and fragmentation. The errors make it difficult to read and understand what ought to be a very interesting and valuable paper.
Line 43: "Which bring challenges to the design of commercial building space traffic flow" is a fragmented sentence. What brings challenges to the design of commercial building space traffic flow? Is it the factors mentioned in the prior sentence such as fuzzy functions and topicalizations? If so, then state that these factors of commercial space use bring challenges and describe some of the challenges.
Line 66: Are the reasons listed based on the consultation with research scholars and planners? If so, do not say the reasons "may be" but that these are the reasons based on the conversations with the research scholars and planners.
Line 70: Explain why a user-centered evaluation system will "probably be" an important step in design. I would also suggest using a different term to "probably be".
Lines 72 - 75: Re-write the sentence
Section '2.1. Commercial building space traffic flow' is well written and explains the gap and goal of the paper clearly. This is what I mean by taking time to proof read the paper so it flows properly, and clearly explains the important story you are trying to tell.
Line 120: As mentioned in the initial review, POE does not include instruments, rather POE+M does include the use of instruments. Please make the distinction and cite POE+M references as needed.
Please proof-read section '2.2. Theoretical Framework', it really needs edits.
Line 406: Remove the additional "4. Results" mentioned. It is subsection 4.2
Line 420: Shouldn't it be sub-subsection 4.2.2?
Line 433: Shouldn't it be subsection 4.3?
Author Response
Please see the attachment.

This manuscript is a resubmission of an earlier submission. The following is a list of the peer review reports and author responses from that submission.
Round 1
Reviewer 1 Report
The topic of the article, the methodology, and the results of the research are definitely interesting.
Title: it is possible to omit the word research to shorten the title, it is usually implied that an article describes a research.
Introduction:
Line 35 “Through expert interviews and field research, it is found that most commercial building space traffic flow design is not ideal after use.” It would be ideal having a reference for this sentence (or at least declare the number of experts and their background that were interviewed, maybe adding in an annex a summary of the interviews, or adding which kind of field studies were performed and a summary or graph of the results). In fact, this sentence represents one of the main reasons for developing the research.
“The most important reason may be the lack of a user demand-centered evaluation 39 system and information feedback method in the design process of commercial building 40 space traffic flow [4].” I kindly recommend checking the reference of this sentence since the topic of the referenced article does not seem to be related to commercial buildings, but to landscape architecture design.
Literature Review:
The literature review should be revised and extended since there are not many references of important authors who developed POEs in the first place such as W.F.E. Preiser (the first researcher to formalize and theorize POEs), A. Leaman, and B. Bordass, and more recent researchers such as W. O’Brien, J. Vischer. In addition, most presented applications are developed in China, while a wider perspective introducing international studies could be useful. Furthermore, all described POE applications could be divided into topics to ease the reading, by describing together in a single paragraph all publications and studies related to one topic, e.g., POE studies aiming at verifying advantages and user opinions regarding refurbished buildings or POEs to verify user satisfaction in office buildings.
Methodology and Results:
The methodology is based on established methods, however the application to verify the design of commercial buildings in relation to user satisfaction represents its innovative character. The methodology steps are clear, however a schema of the methodology and application steps could be useful to better clarify the process.
The quality of the graphs of the results may be improved.
Finally, the results and conclusions seem sound, however the English of the whole manuscript needs to be improved and revised for the final manuscript version.
Reviewer 2 Report
The paper presents a traffic flow evaluation index system then applies it in a specific case study to determine users satisfaction. The contribution of the paper, in my opinion lies in the development of the evaluation index system for traffic flow. However, it lacks scientific rigor, at least in the format presented, on the process and its validity in its construction or development. This section will benefit to be elaborated further.
The application of these evaluation indexes to a case study yields some information as an applied process. However, the design recommendations as derived from the results are case-specific or very generic and may not be generalized, a limitation acknowledged in the conclusion, thus may be of limited relevance to a broader readership working on the same topic.
Below are some general and specific notes, comments and/or recommendations and items for consideration.
- The title should reflect / include the AHP method in this inquiry as it forms an important contribution to the paper rather than reading as an application case study of the method. The word “Research” might be omitted.
- It is recommended to include the aim and objectives of the present study in the introduction part
- It is not clear how the literature review related to energy savings in commercial buildings relates to the scope of the paper (lines 44 to 70). If the point is to state that research on commercial buildings has primarily focused on energy efficiency and lighting then it may be summarized and supported by references.
- Similar comment regarding POE as an investigative tool, the examples cited span though a range that are neither fully comprehensive nor directly related to the scope, and are primarily descriptive and lack a critical evaluation as to what is derived that fits the aim of the present Examples in context may have some relevance but most of the reported studies do not relate to the typology under consideration within or beyond China.
- It is also recommended to add in the literature part, the definition and parameters of traffic flow design with a focus on specific characteristics in commercial buildings so as to serve and support the POE in the case study.
- It is also recommended that the Evaluation indexes should be introduced and defined too.
Methodology
- Section 3.2; should probably come first under the methodology and be more explicit regarding the interviews and questionnaire referred
- Line 192: the “various indicator” should be more explicit
- The rationale for the development of the Traffic flow satisfaction Criterion/factors and level as illustrated in Figure 4 should be better explained. This is an important step in the paper but is not detailed.
The paper will benefit from full proofreading editing and linguistic review. Below are some examples that need to be addressed;
- Some paragraphs are rather confusing and may gain in clarity with shorter paragraphs – for example, paragraphs in lines 169-173
- Line 127 – unclear meaning “Sea aided navigation was also optimized after Jinbiao Chen's [29] research using Poe”
Round 2
Reviewer 2 Report
The manuscript has been significantly improved.
I recommend reviewing the conclusion in terms of writing style (avoid long sentences and informal writing) while including the main findings/results along with the stated limitations.
